# Channel State Information Based Indoor Fingerprinting Localization

**DOI:** 10.3390/s23135830

**Published:** 2023-06-22

**Authors:** Rongjie Che, Honglong Chen

**Affiliations:** 1Sinopec Petroleum Engineering Corporation, Dongying 257026, China; cherj.osec@sinopec.com; 2College of Control Science and Engineering, China University of Petroleum (East China), Qingdao 266580, China

**Keywords:** indoor localization, channel state information, fingerprinting-based, IWKNN, full-dimensional subcarriers

## Abstract

Indoor localization is one of the key techniques for location-based services (LBSs), which play a significant role in applications in confined spaces, such as tunnels and mines. To achieve indoor localization in confined spaces, the channel state information (CSI) of WiFi can be selected as a feature to distinguish locations due to its fine-grained characteristics compared with the received signal strength (RSS). In this paper, two indoor localization approaches based on CSI fingerprinting were designed: amplitude-of-CSI-based indoor fingerprinting localization (AmpFi) and full-dimensional CSI-based indoor fingerprinting localization (FuFi). AmpFi adopts the amplitude of the CSI as the localization fingerprint in the offline phase, and in the online phase, the improved weighted K-nearest neighbor (IWKNN) is proposed to estimate the unknown locations. Based on AmpFi, FuFi is proposed, which considers all of the subcarriers in the MIMO system as the independent features and adopts the normalized amplitudes of the full-dimensional subcarriers as the fingerprint. AmpFi and FuFi were implemented on a commercial network interface card (NIC), where FuFi outperformed several other typical fingerprinting-based indoor localization approaches.

## 1. Introduction

Location information is indispensable in the age of the Internet of Things (IoT) [1,2,3,4,5,6,7,8,9,10]. In many applications in confined spaces [11,12], such as tunnels, mines, and culverts, a precise location service is required to perform necessary operations for different kinds of projects. The Global Positioning System (GPS) is widely used to provide outdoor location information, providing a great convenience to travel. Unfortunately, GPS is unable to complete accurate indoor localization in confined spaces due to factors such as mutual occlusion between buildings and the complexity of the indoor environment. However, indoor location information has a high economic value. Therefore, indoor localization is of great significance and has attracted much attention from the research community.

In the literature, many different wireless communication standards have been applied for indoor localization, such as WiFi [13], RFID [14], ZigBee [15], Bluetooth [16], ultrasonic [17], etc. Among them, WiFi has attracted much attention due to its low cost, simple deployment, and ubiquity.

The approaches using WiFi for localization can be classified into two categories: link-based and fingerprinting-based. Link-based approaches estimate the location of the target by constructing a model of the WiFi signal transmission. The commonly used models are the time of arrival (ToA) [18], round trip time (RTT) [19], time difference of arrival (TDoA) [20], angle of arrival (AoA) [21], etc. However, it is difficult to establish an accurate model due to the complex multipath effect and noise interference.

On the other hand, fingerprinting-based localization approaches that do not require mathematical model are less affected by multipath and can achieve good indoor localization results. The fingerprinting-based approaches consist of two phases: the offline phase and the online phase. In the offline phase, the locations and their associated features are correlated to construct a fingerprint database. In the online phase, the features of the target are mapped to the fingerprint database to estimate its location.

Fingerprinting-based indoor localization systems using WiFi typically collect received signal strength (RSS) or channel state information (CSI) as the features to construct the fingerprint database. The RSS has been widely used in fingerprinting-based localization approaches for more than a decade. Radar [22] and Horus [23] are two of the most-famous approaches to use the RSS for indoor localization. However, the RSS has poor robustness and weak ability to distinguish the target’s location, so the RSS is not an ideal feature for localization. Therefore, more-stable and finer-grained information is required for accurate indoor localization.

In IEEE 802.11n, where the orthogonal frequency division multiplexing (OFDM) and multiple input multiple output (MIMO) technologies are applied, the wireless communication link between the transmitter and the receiver can be quantified and evaluated using the CSI. The CSI can fully reflect the multipath effect of the physical environment, which is more fine-grained than the RSS. In addition, mainstream vendors of commercial WiFi such as Intel have gradually opened up the physical layer information, making it convenient to obtain the CSI of WiFi with open-source tools. FILA [24] claimed that it is the first technology to apply the CSI to indoor localization. Using the acquired CSI, FILA constructs a model of indoor signal propagation for localization. Compared with the approaches of using the RSS, FILA greatly improves the localization accuracy, with an average accuracy at the sub-meter level. Then, FIFS [25], the first approach combining the CSI and fingerprinting-based approach, emerged, and its localization accuracy is higher than that of Horus.

In this paper, based on the CSI, two fingerprinting-based indoor localization approaches are proposed, namely AmpFi and FuFi. AmpFi uses the amplitudes of the collected subcarriers as the features to construct the fingerprint database in the offline phase. In the online phase, to mitigate the effect of outliers in reference points on localization, an improved weighted K-nearest neighbor (IWKNN) algorithm is proposed to estimate the location of the target. To improve the performance, based on AmpFi, FuFi makes full use of the rich information in the MIMO system and creatively uses the full-dimensional subcarriers as independent variables to achieve high-precision localization.

The main contributions of this paper are the following:A fingerprinting-based localization method, AmpFi, is proposed, which uses the amplitude as the fingerprint and estimates the locations of the target using IWKNN.Based on AmpFi, FuFi is proposed, which fully takes into account the variability of subcarriers among multiple antennas in the MIMO system and adopts the normalized amplitudes of full-dimensional subcarriers as the fingerprint.Both AmpFi and FuFi were implemented in a commercial network interface card (NIC), and their performances were evaluated in two different environments, a living room and a conference room. The experimental results showed that FuFi outperformed AmpFi and several other typical fingerprinting-based indoor localization approaches.

The rest of this paper is organized as follows. Section 2 reviews the related work. Section 3 presents some preliminaries. In Section 4, the architectures of FuFi and AmpFi are described in detail. In Section 5, FuFi, AmpFi, and four other fingerprinting-based indoor localization approaches are implemented in two scenarios: a living room and a conference room, and their performance is evaluated. Finally, we conclude this paper in Section 6.

## 2. Related Work

The Linux 802.11n CSI Tool [26], which allows obtaining the CSI on a commercial NIC, inspired the use of the CSI for indoor localization. Previously, the CSI was only available through specific signal meters. FILA [24] claimed that it was the first method for indoor localization using the CSI, and it achieved sub-meter-level localization by constructing a model of signal propagation. At the same time, the first approach to integrate the CSI with fingerprinting-based localization, FIFS [25], was released, which took full advantage of the spatial diversity and frequency diversity of the CSI, and its localization accuracy exceeded that of approaches using the RSS. PinLoc designed by Sen et al. [27] transformed the localization problem into a classification problem, and it was performed by a machine learning approach and achieved an average classification accuracy of 89% in the experiments. CSI-MIMO [28] utilized multiple antennas in a MIMO system to create a new fingerprint by calculating the difference between the amplitudes and phases of adjacent subcarriers, and its localization error was reduced by 57% compared to FIFS. Tian et al. [29] adopted a different idea. They proposed to extract the CSI by K-means clustering to reflect the influence of the multipath effect on localization, and its localization accuracy was improved by 24% compared with CSI-MIMO in the case of a single AP.

As research progressed, deep learning was applied to indoor localization. Wang et al. [30] proposed DeepFi. In the offline phase, deep learning was used to train the weight of the amplitude as the fingerprint, and in the online phase, the radial basis function was used to locate the target. In addition to DeepFi, they also proposed PhaseFi [31], which obtained more accurate phase information of the CSI by linear transformation to replace the amplitudes of subcarriers and also obtained decent localization results. Chang et al. [32] enabled a single AP, non-intrusive indoor localization with deep neural networks. Singular-value decomposition was used to pre-process for the CSI, and noise injection and an interpolation-based approach were used for data enhancement. They achieved sub-meter-level localization. Zheng et al. [33] designed a self-calibrating time-reversal fingerprinting localization method for the problem of different environments in offline and online phases and updated the fingerprint database in real-time by deep learning to mitigate the impact of environmental changes on localization. Berruet et al. [34] switched the target to address the problem of the high complexity of CSI data for exhaustive deep learning. They considered different data acquisition scenarios with multiple antenna elements on the gateway and demonstrated that kernel entropy component analysis had better performance for data reduction and was more beneficial for indoor localization. Zhang et al. [35] achieved a localization accuracy of about 0.9 m in open and complex indoor environments by optimizing the ratio of the amplitude and phase. In the offline phase, the amplitude and phase of different proportions were selected to form a dataset, and the dataset was trained with LSTM to obtain the model. In the online phase, the target’s location was predicted from the obtained model by regression.

In addition to the above, there are some interesting studies on indoor localization with WiFi. The goal of Chen et al. [36] was to solve the problem that the maximum bandwidth of WiFi is only 40 MHZ and the multipath effect in the environment cannot be effectively solved. The bandwidth was expanded through frequency hopping, and the location fingerprint was synthesized from the average of the CSI of different channels. In the online phase, the localization of the target was achieved with time-reversal resonating strength, and centimeter-level localization accuracy was obtained in their experimental environment. Zhang et al. [10] designed a method to achieve more accurate localization with a single AP for certain places where robust WiFi signals are not available. A new phase decomposition method was proposed to obtain the true phase, and principal component analysis was used to reduce the dimensionality of the fingerprint. Finally, they proved that their approach outperformed Radar [22] and CSI-MIMO [28]. Chen et al. [37] combined link-based and fingerprinting-based approaches for indoor localization. In the offline phase, the autoregressive modeling entropy of the CSI amplitude was adopted as the location fingerprint and the angle of arrival (AoA) method was used to correct the fingerprint to construct a reliable database. In the online phase, a new binary kernel regression scheme was proposed to achieve localization.

The previous works did not pay attention to the variability between different antenna pairs in MIMO systems. In this paper, we took full advantage of this variability to achieve highly accurate indoor localization. The CSI of all antenna pairs was treated as independent features, and the full-dimensional CSI was normalized as the location fingerprint. A new fingerprint database was constructed, and eventually, excellent localization results were achieved.

## 3. Preliminaries

In this section, we first introduce the OFDM and MIMO techniques, which are widely used. After that, the CSI and its characteristics are described.

### 3.1. Orthogonal Frequency Division Multiplexing

OFDM is a multi-carrier digital modulation technology, which splits the high-rate information data stream into multiple low-rate parallel data streams through serial/parallel (S/P) conversion. OFDM modulates each low-rate data stream using the independent carrier and superimposes them together to form a transmitting signal.

The working process of the OFDM system is shown in Figure 1. At the OFDM transmitter, the serial input data are converted into N parallel data and assigned to N different subcarriers. These data are passed through the channel in a serial manner by the inverse fast Fourier transform (IFFT) and parallel/serial (P/S) conversion. After the digital-to-analog converter (DAC) and low-pass filter, the OFDM signals are finally output. The reception process of OFDM signals is the inverse of the transmitting process. The received signals are converted into the frequency domain after the fast Fourier transform (FFT) and, finally, recovered to binary data at the transmitter after P/S conversion and decoding.

In signal propagation, different subcarriers will have different attenuations due to interference, resulting in unique amplitude and phase information for each subcarrier. The amplitudes and phases of these subcarriers can be used for indoor localization.

### 3.2. Multiple-Input Multiple-Output

MIMO technology refers to the use of multiple transmitting and receiving antennas to transmit and receive signals, respectively, thus improving communication quality. MIMO systems multiply the system channel capacity without increasing the spectrum resources and antenna transmission power, showing obvious advantages.

MIMO is widely used in WLAN, and there are many types of network equipment with 2 × 2, 3 × 3, and even more antennas, greatly improving the efficiency of data transmission.

### 3.3. Channel State Information

In wireless communication, the CSI represents the channel quality of the communication link. It describes the attenuating factors on each communication link, such as scattering, distance, and interference from other signals. Thus, the CSI is more fine-grained than the RSS. In a narrowband flat-fading channel, the CSI can be represented in the frequency domain as follows:(1)Y=HX+ζ,
where *Y* is the set of received vectors, *X* is the set of sent vectors, and *H* and ζ are the channel matrix and noise matrix, respectively.

The channel matrix *H* is composed of the physical layer information of the *N* subcarriers and can be expressed as follows:(2)$$H={\left[H_{1},H_{2},H_{3},\ldots ,H_{N}\right]}^{T}.$$

For a single-input single-output (SISO) system, $H_{i}$ is the CSI of the i-th subcarrier and is denoted as follows:(3)$$H_{i}=\left|H_{i}\right|e^{j∠H_{i}},$$
where $\left|H_{i}\right|$ and $∠H_{i}$ are the amplitude and phase of the i-th subcarrier, respectively.

For a MIMO system with p transmitting antennas and q receiving antennas, $H_{i}$ can be expressed as follows:(4)$$H_{i}=\left(\begin{array}{cccc} h_{i}^{11} & h_{i}^{12} & \cdots & h_{i}^{1q}\\ h_{i}^{21} & h_{i}^{22} & \cdots & h_{i}^{2q}\\ \vdots & \vdots & \ddots & \vdots \\ h_{i}^{p1} & h_{i}^{p2} & \cdots & h_{i}^{pq} \end{array}\right).$$

In the above system, the channel matrix *H* has a *N* × *p* × *q* dimension, where $h_{i}^{pq}$ represents the amplitude and phase information of the *i*-th subcarrier between the *p*-th transmitting antenna and the *q*-th receiving antenna. Each $h_{i}$ can be expressed as in Equation (Equation 3).

## 4. Full-Dimensional CSI-Based Localization Approach

In this section, the working processes of AmpFi and FuFi are presented, and they both use IWKNN to estimate the location of the target. IWKNN is an improved algorithm based on the weighted K-nearest neighbor (WKNN) [38]. WKNN uses the locations of the K reference points that are most similar to the test point to represent the location of the test point, and WKNN does not care about the location distribution of these K reference points. In contrast, IWKNN takes into account that these K nearest neighbors may contain outlier points that are significantly far from other nearest neighbors. The presence of outliers can lead to degradation for localization accuracy, and IWKNN can effectively reject these outliers.

The system overview for these two approaches is shown in Figure 2, which consists of two phases: the offline phase and the online phase. In AmpFi and FuFi, the first two steps of the offline phase are the same as the online phase, in which it is required to collect and process the raw CSI data. In the offline phase, the data are stored in a fingerprint database according to the respective requirements. In the online phase, the processed data are converted into a specific format and compared with the information stored in the fingerprint database to locate the target.

FuFi and AmpFi are composed of four parts: data collection, data processing, establishment of the fingerprint database, and location estimation. They are different only in the last two parts, and these four parts will be detailed below.

### 4.1. Data Collection

Collecting stable and reliable data is the first step in a fingerprinting-based approach, and both the offline and online phases start with data collection. In our FuFi and AmpFi, the mobile device was placed in a fixed location, and the user stood in different locations, with changes in the location of the user resulting in the different CSI received by the mobile device. In the offline phase, the user was asked to stand at different locations known to us, which we call reference points (RPs). The CSI was recorded when the user stood at the RP, along with the location of the RP. In most cases, only information about the location in a two-dimensional plane was required, so only two-dimensional information of the location was recorded. Thus, the location of an RP $rp_{a}$ can be recorded as ($rp_{a,x}$, $rp_{a,y}$). In the online phase, the user could stand in any location unknown to us, which we call the test point (TP), and the CSI was recorded. The number of data to be collected in each point was not fixed, and in this paper, it was set to 500 and 1000. The training data and the test data were independent. The locations of the reference points were fixed, while the locations of the test points were chosen randomly. All reference points were first sampled, followed by the test points, with a large time interval between them.

### 4.2. Data Processing

Direct use of the raw CSI can lead to an error in localization due to the effect of interference, so the data must be processed. In our approaches, the data processing consisted of two parts: amplitude calculation and outlier rejection.

#### 4.2.1. Amplitude Calculation

Some reasons make the original phase of the CSI subject to error and cannot be used as the feature to distinguish locations, such as the lack of clock synchronization between the transmitter and receiver, the mismatch between transmitter and receiver frequencies, and the noise interference. The amplitude of the CSI is more stable than the phase in a stationary environment, so the amplitude was chosen as the key feature to distinguish locations. The CSI that is usually collected directly does not consist of the amplitude and phase as shown in Equation (Equation 3), but is in the form of a complex number such as a+bi. Therefore, the amplitude needs to be calculated, and the formula is shown in Equation (Equation 5).
(5)$$\left|h\right|=\sqrt{a^{2}+b^{2}},$$
where $\left|h\right|$ is the absolute value of the amplitude of this subcarrier.

#### 4.2.2. Outlier Rejection

Outliers are values that significantly differ from other values in the dataset. Inevitably, our measurement process was subject to human or environmental interference, which could lead to outliers. The introduction of outliers had an adverse effect on our subsequent localization, so the outliers needed to be removed to reduce the localization error. The PauTa criterion can eliminate outliers and reduce gross error in the measurement process.

The PauTa criterion assumes that a set of test data contains only random error, and the data that differ from the mean by more than three-times the standard deviation are considered to contain gross error, so these data should be rejected.

During our data collection process, the CSI of each location was measured several times, and using this criterion, highly abnormal outliers could be removed, after which, the average of the data needed to be calculated again to obtain the true amplitude.

### 4.3. Establishment of Fingerprint Database

Data collection and data processing are parts of both the offline and online phases, and the establishment of the fingerprint database was one of the parts in the offline phase. In this part, a correspondence between the CSI and the location needed to be established so that the locations of the test points could be located based on the fingerprint database during the online phase. It was also in this part that AmpFi and FuFi were different, and the differences will be described in detail below.

#### 4.3.1. AmpFi

In a MIMO system with *p* transmitting antennas and *q* receiving antennas, *p* × *q* antenna pairs can be formed. In the above system, the amplitude of *i*-th subcarrier after data processing is a matrix of *p* × *q* as follows:(6)$$\left(\begin{array}{cccc} \left|h_{i}^{11}\right| & \left|h_{i}^{12}\right| & \cdots & \left|h_{i}^{1q}\right|\\ \left|h_{i}^{21}\right| & \left|h_{i}^{22}\right| & \cdots & \left|h_{i}^{2q}\right|\\ \vdots & \vdots & \ddots & \vdots \\ \left|h_{i}^{p1}\right| & \left|h_{i}^{p2}\right| & \cdots & \left|h_{i}^{pq}\right| \end{array}\right).$$

In AmpFi, $F_{i}$ is used to denote the amplitude of the *i*-th subcarrier and $F_{i}$ is calculated as follows:(7)$$F_{i}=\frac{1}{pq}\sum_{n=1}^{p}\sum_{m=1}^{q}|h_{i}^{nm}|.$$

Using $rp_{a}$ as an example, the fingerprint of $rp_{a}$ should be recorded in AmpFi’s fingerprint database as follows:(8)$$\left(\begin{array}{c} F_{1},F_{2},F_{3},\cdots ,F_{N},rp_{a,x},rp_{a,y} \end{array}\right),$$
the dimension of this row vector is *N* + 2.

#### 4.3.2. FuFi

Unlike AmpFi, FuFi focuses on the differences in the same subcarrier between multiple antennas in the MIMO system. The Linux 802.11n CSI Tool was used to collect the CSI, which was based on the Intel WiFi Wireless Link 5300 802.11n MIMO radios and can read 30 subcarriers. From the data collected, we found that the CSI amplitudes differed considerably between multiple antenna pairs for the same subcarrier. As shown in Figure 3, the amplitude of Antenna A was between 5 dB and 15 dB, while that of Antenna B was between 20 dB and 33 dB; the subcarriers’ amplitudes of Antenna Pair *A* and Antenna Pair *B* were very different, so it was not reasonable to average the values of multiple antenna pairs when calculating the amplitudes of the subcarriers. Therefore, FuFi treated the subcarriers of all antenna pairs as independent features, which can fully utilize the full-dimensional CSI of the MIMO system to improve the localization accuracy.

In addition, we found that the range of amplitude transformation that resulted from the location changing significantly differed between different antenna pairs. When the user stood at two locations LA and LB, which were 1 m away from each other, the amplitude changes of multiple antenna pairs were measured, respectively. The experimental results are shown in Figure 4. The range of amplitude variation for Antenna Pair B was between 0.15 dB and 1.89 dB, while the range of variation for Antenna Pair A was much smaller, i.e., between 0 dB and 0.37 dB. Therefore, the effect of Antenna Pair B on localization was much higher than that of Antenna Pair A.

In order to make full use of the amplitude of each antenna pair, FuFi normalizes the measured amplitude using Equation (Equation 9) as follows:(9)$$\left|h_{i}^{*}\right|=\frac{\left|h_{i}\right|-\left|h_{i}^{min}\right|}{\left|h_{i}^{max}\right|-\left|h_{i}^{min}\right|},$$
where $\left|h_{i}\right|$ is the measured amplitude of the *i*-th subcarrier in the antenna pair being normalized, $\left|h_{i}^{max}\right|$ is the maximum amplitude of the *i*-th subcarrier in all reference points in this antenna pair, and similarly, $\left|h_{i}^{min}\right|$ is the minimum value, while $\left|h_{i}^{*}\right|$ is the amplitude of the *i*-th subcarrier in this antenna pair after being normalized. Normalization was performed for each subcarrier of each antenna pair individually. FuFi maps the amplitudes to a value between 0 and 1, and each antenna pair can participate fairly in the location estimation.

Similar to AmpFi, in FuFi, $F_{i}$ is also used to denote the amplitude of the i-th subcarriers in the MIMO system, with $F_{i}$ denoted as follows:(10)$$F_{i}=\left(\begin{array}{c} \left|h_{i}^{*11}\right|,\left|h_{i}^{*12}\right|,\cdots ,\left|h_{i}^{*1q}\right|,\left|h_{i}^{*21}\right|,\cdots ,\left|h_{i}^{*pq}\right| \end{array}\right).$$

The fingerprint of FuFi at the location of $rp_{a}$ can be similarly expressed as Equation (Equation 8), but the difference is that the row vector dimension is *N* × *p* × *q* + 2. For systems with more than one AP, the fingerprint database for all APs needs to be constructed separately.

### 4.4. Location Estimation

In the online phase, the different online CSI matrix needs to be formed to fit different approaches. The data collected in the online phase undergoes the same processing as in the offline phase, during which the online CSI matrix as shown in Equation (Equation 11) will be generated. Note that $F_{i}$ can be calculated using Equation (Equation 7) for AmpFi and Equation (Equation 10) for FuFi.
(11)$$\left(\begin{array}{c} F_{1},F_{2},F_{3},\cdots ,F_{N} \end{array}\right).$$

In particular, during the amplitude normalization in FuFi, the amplitude of a test point was set to 1 if it was greater than the amplitudes of all the reference points in the same subcarrier of the same antenna pair, and it was set to 0 if it was smaller than the amplitudes of all the reference points.

After obtaining the online CSI matrix, both AmpFi and FuFi use the IWKNN algorithm for location estimation. The steps of IWKNN are as follows.

#### 4.4.1. Calculating Euclidean Distance

The Euclidean distance was used to express the difference between the amplitudes of the test point and the reference point. The calculation of the Euclidean distance is illustrated using reference point $rp_{a}$ and test point $tp_{b}$ as examples. For AmpFi, the Euclidean distance is calculated as follows:(12)$$dis\left(rp_{a},tp_{b}\right)=\sqrt{\sum_{i=1}^{N}{\left(F_{i,rp_{a}}-F_{i,tp_{b}}\right)}^{2}},$$
where $F_{i,rp_{a}}$ and $F_{i,tp_{b}}$ are the amplitudes of the *i*-th subcarrier when the target is located at $rp_{a}$ and $tp_{b}$, respectively.

For FuFi, the Euclidean distance is calculated as follows:(13)$$dis\left(rp_{a},tp_{b}\right)=\sqrt{\sum_{i=1}^{N}\sum_{j=1}^{p}\sum_{k=1}^{q}{\left({\left|h_{i}^{*jk}\right|}_{rp_{a}}-{\left|h_{i}^{*jk}\right|}_{tp_{b}}\right)}^{2}},$$
where ${\left|h_{i}^{*jk}\right|}_{rp_{a}}$ and ${\left|h_{i}^{*jk}\right|}_{tp_{b}}$ are the normalized amplitudes of the *i*-th subcarrier between the *j*-th transmitting antenna and the *k*-th receiving antenna when the target is located at $rp_{a}$ and $tp_{b}$, respectively.

If there is more than one AP in the system, the distance between these two points for each AP needs to be calculated separately and then summed up as the final result.

#### 4.4.2. Selecting Nearest Neighbors

After obtaining all the Euclidean distances between the reference points and test point, the *K* reference points with the smallest Euclidean distance to the test point were selected as K nearest neighbors. Then, the average coordinates of these K nearest neighbors were calculated and denoted as $\left(M_{x},M_{y}\right)$. After that, a distance threshold $T_{dis}$ was set, which was a constant. Note that the values of K and $T_{dis}$ were obtained by extensive experiments in labs. To optimize the localization performance, the optimal values of K and $T_{dis}$ in the current indoor environment were obtained by extensive experiments in the offline phase.

#### 4.4.3. Rejecting Outliers in Reference Points

The distances from the K nearest neighbors to $\left(M_{x},M_{y}\right)$ were calculated separately, and the shortest distance $d_{min}$ and the farthest distance $d_{max}$ among them were found. If $d_{max}>d_{min}+T_{dis}$, the farthest reference point from $\left(M_{x},M_{y}\right)$ in the K nearest neighbor was considered as an outlier and was rejected in the localization of this test point. Then, the value of K was subtracted by 1, and $\left(M_{x},M_{y}\right)$ was recalculated. If $d_{max}\leq d_{min}+T_{dis}$, it was considered that there were no outliers in the K nearest neighbors.

This step was repeated until there were no outliers in the K nearest neighbors.

#### 4.4.4. Estimating the Location of the Test Point

After obtaining the K nearest neighbors without outliers, Equation (Equation 14) was used to calculate the coordinates $\left(tp_{x},tp_{y}\right)$ of the test point:(14)$$\left(tp_{x},tp_{y}\right)=\sum_{k=1}^{K}\left(\frac{\frac{1}{dis(rp_{k},tp)+\epsilon }}{\sum_{k=1}^{K}\frac{1}{dis(rp_{k},tp)+\epsilon }}\cdot \left(rp_{k,x},rp_{k,y}\right)\right),$$
where $dis(rp_{k},tp)$ is the Euclidean distance between the *k*-th nearest neighbor and the test point, ($rp_{k,x},rp_{k,y})$ is the localization coordinate of the *k*-th nearest neighbor, and ϵ is a very small constant, and in this paper, we took ϵ=0.0001 to prevent the denominator from being equal to 0.

## 5. Performance Evaluation

In this section, our experimental scenarios are first introduced. The experiments were conducted with two different scenarios: a conference room and a living room. After that, the performances of our FuFi and AmpFi with other fingerprinting-based localization approaches were compared, and the same reference points and test points were used for a fair comparison. Finally, the effects of some factors on the performance of the localization approaches were analyzed, such as the number of APs, the number of reference points, etc. Specifically, three APs and a laptop were placed in fixed positions in our labs when collecting the data. In the offline phase, the users were asked to stand at the reference points successively, and the CSI was collected when the users stood at different reference points. In the online stage, the user stood at the test point randomly, and the CSI of the user was collected for positioning.

### 5.1. Experimental Scenarios

The experiments were conducted in two different scenarios, and $d_{x}$ and $d_{y}$ are used to represent the horizontal and longitudinal distances between two adjacent reference points.

#### Conference Room and Living Room

First, a 8.5 m × 7.5 m conference room was used as the experimental site. Three APs were deployed at the specific locations in the conference room, which is shown in Figure 5. Both $d_{x}$ and $d_{y}$ were equal to 1 m. In the offline phase, the CSI was collected from 49 reference points to construct a fingerprint database. The data were collected 1000 times per location to improve the accuracy of the data. The sampling interval was set to 0.01 s.

Second, the experiment was conducted in a living room, which was 6 m × 7 m and also covered by three APs. In this scenario, $d_{x}$ and $d_{y}$ were set to 0.9 m. The fingerprint was collected at 36 different reference points as shown in Figure 6. The data were collected 500 times per location, and the sampling interval was set to 0.01 s.

The experimental system is shown in Figure 7, which consisted of a Dell laptop equipped with an Intel 5300 wireless network card as the mobile device and three TL-WR886N routers, which work at a bandwidth of 20 MHZ, as the APs. In our experiments, each AP used two transmitting antennas, and the mobile device used three receiving antennas. In order to reduce the error, the user was asked to stand at different locations using the same posture.

### 5.2. Performance Evaluation

The mean distance error (MDE) was used as a criterion to evaluate the superiority of localization approaches. First, the MDEs of FuFi and AmpFi with four other fingerprinting-based localization approaches were evaluated in the single-AP case, including FIFS [25], CSI-MIMO [28], CSI indoor localization based on K-means clustering [29] (hereafter referred to as K-means), and FUSELOC [39].

The results of the experiment conducted in the conference room are shown in Figure 8a. We can find that FuFi, which uses full-dimensional CSI information, had the smallest MDE, i.e., 0.65 m. The MDEs of FIFS, CSI-MIMO, K-means, FUSELOC, and AmpFi were 1.62 m, 1.10 m, 0.74 m, 1.34 m, and 1.43 m, and the gains of FuFi compared to them were 59.9%, 40.9%, 13.8%, 51.5%, and 54.5%, respectively. The results showed that the performance of FuFi was significantly better than the other fingerprinting-based localization approaches. To improve the credibility of our conclusion, the experiment was repeated with an AP in the living room. The experimental results are shown in Figure 8b, where we can also find that FuFi outperformed the other approaches. In the living room, the gains of FuFi compared to FIFS, CSI-MIMO, K-means, FUSELOC, and AmpFi were 28.2%, 24%, 13.3%, 16.4%, and 19.3%, respectively. The gains were reduced, and we speculated that the cause may be the more complex environment in the living room and interference from other wireless signals. In this scenario, the accuracy of all localization approaches decreased, but the performance of FuFi was still the best.

The cumulative distribution function (CDF) of the distance error is also a measure to compare the performances of localization approaches. Figure 9a shows the CDFs of the distance error in the conference room with a single AP. In the experiment, the CDF of FuFi was the first to reach 1, and over 90% of the test points had distance errors of 1.2 m or less. When the CDFs of FIFS, CSI-MIMO, K-means, FUSELOC, and AmpFi exceeded 90%, the distance errors were 2.45 m, 2.03 m, 1.5 m, 3.1 m, and 3.8 m, respectively. Similarly, the CDFs of the distance error in the living room are shown in Figure 9b. The results showed that the CDF of FuFi was still the first approach to reach 1, and more than 90% of the test points had distance errors within 2.2 m. In terms of the CDFs of the distance error, FuFi outperformed the other fingerprinting-based localization approaches.

To further show the distribution of the distance errors for each localization method, we plot the respective box plots for the experimental results. Figure 10a shows the box plot in the conference room. It can be found that the upper quartile, median, and lower quartile of the distance error of FuFi were the smallest. It can be concluded that FuFi’s ability to locate most of the test points was better than several other methods. The box plot of the experiment in the living room is shown in Figure 10b; similarly, it can be found that FuFi still has the smallest upper, median, and lower quartiles of the distance error. Therefore, we can conclude that FuFi is the best-performing localization approach.

In addition, the effect of the number of reference points on the localization performance was explored in the same experimental region. In the conference room, the number of reference points in the region was modified to 28 and 16. Correspondingly, $d_{x}$ was set as 1 m, 2 m, and $d_{y}$ was set as 2 m, 2 m, respectively. The layout of the reference points is shown in Figure 11.

The experimental results are shown in Figure 12a, which shows that the performances of all fingerprinting-based localization approaches deteriorated as the number of reference points decreased. The MDE of FuFi changed from 0.65 m to 1.33 m and 1.63 m, respectively, but even so, the performance of FuFi was still the best.

The same comparison experiment was performed in the living room. The number of reference points was modified to 18 when $d_{x}$ was set as 1.8 m and $d_{y}$ was equal to 0.9 m. The layout of the reference points is shown in Figure 13. The experimental results are shown in Figure 12b. The following conclusions can be obtained: the performances of the fingerprinting-based localization approaches gradually deteriorated with the decrease of the number of reference points; under the same conditions, the localization accuracy of FuFi was higher than that of other fingerprinting-based localization approaches.

After that, our conclusions were verified using the CDF of the distance error. FuFi was selected to compare with the previously most-prominent approach K-means, and the experimental results in the conference room and living room are presented in Figure 14a,b, respectively. Similarly, FuFi outperformed K-means, and their performances deteriorated as the number of reference points decreased.

Therefore, in the same area, the localization performance can be improved by setting more reference points, but it will also greatly increase the workload of the fingerprint database construction and the computational complexity of the online phase. Thus, in practical applications, a trade-off should be made to achieve a balance between the workload and localization accuracy.

Finally, the effect of the number of APs on the localization accuracy was explored. In the conference room, both $d_{x}$ and $d_{y}$ were equal to 1 m, and in the living room, $d_{x}$ and $d_{y}$ were set as 0.9 m. In two typical scenarios, the CSI of all three APs was collected separately with the mobile device. We use *n* to represent the number of APs used in a single experiment. Since K-means and FUSELOC do not provide localization strategies when there are multiple APs, the K-means and FUSELOC were not included in this experiment. Figure 15 shows the MDEs of four localization approaches when the number of APs was different. In Figure 15a, the change in the number of APs did not have a significant impact on the localization accuracy, and the amount of change in the MDEs for each approach was within 0.2 m. The experimental results in the living room are shown in Figure 15b. Similarly, the number of APs did not severely affect the localization accuracy, and FuFi still achieved the best performance.

## 6. Conclusions

In this paper, an online localization algorithm, IWKNN, was designed, which can reject outliers in K nearest neighbors and achieve effective localization. Then, AmpFi was proposed, which uses the amplitude of the CSI as the fingerprint and IWKNN to estimate the location of the target. After that, based on AmpFi, FuFi was proposed by using the normalized amplitudes of full-dimensional subcarriers as the fingerprint. Finally, FuFi and AmpFi were implemented using a commercial NIC. To demonstrate the advantages of our approaches, experiments were conducted in a conference room and a living room, respectively, and the results showed that FuFi outperformed AmpFi and four other fingerprinting-based location approaches. In addition, the experimental results showed that the number of APs did not significantly affect the localization accuracy when the WiFi signal was strong enough in the localization area; the larger the intervals between adjacent reference points were, the worse the localization performance was. In the experiments, FuFi achieved the best localization performance under difference scenarios.

## Figures and Tables

**Figure 1 sensors-23-05830-f001:**
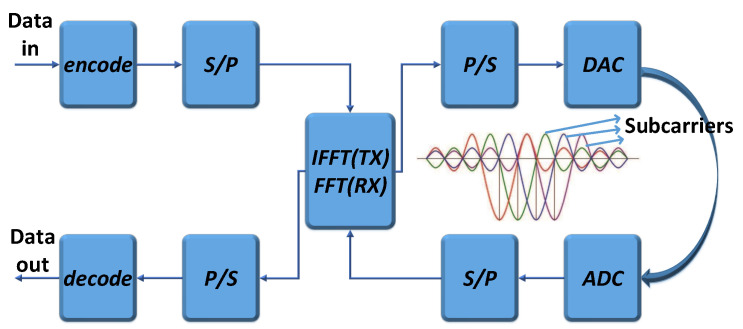
Working process of OFDM transceiver.

**Figure 2 sensors-23-05830-f002:**
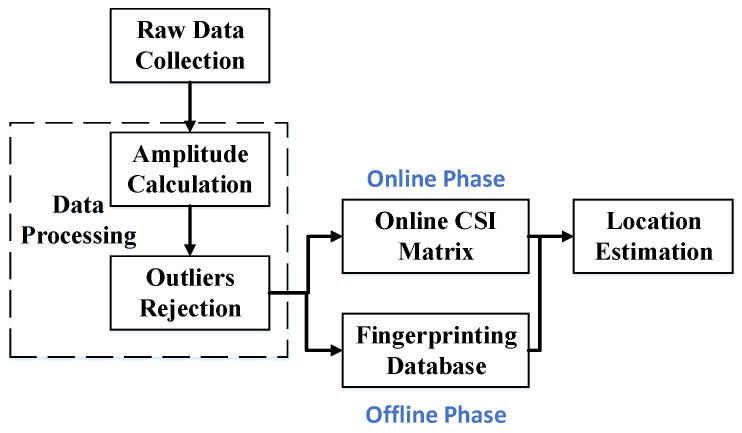
System overview for FuFi and AmpFi.

**Figure 3 sensors-23-05830-f003:**
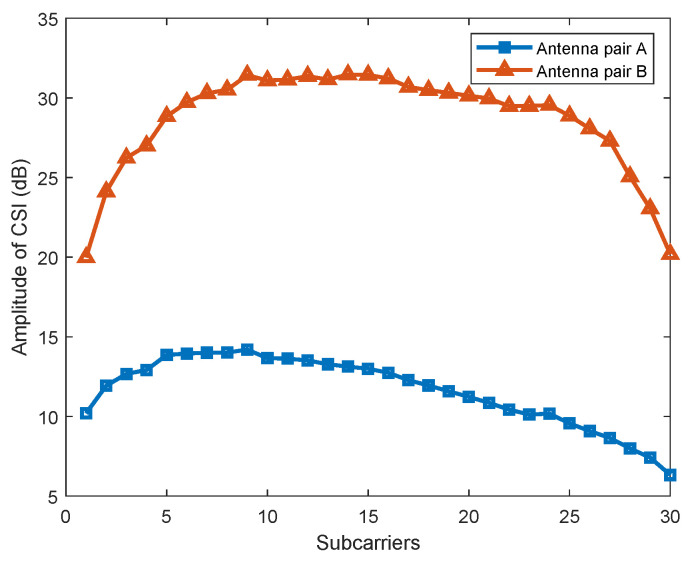
Amplitude of the CSI for different antenna pairs in MIMO systems.

**Figure 4 sensors-23-05830-f004:**
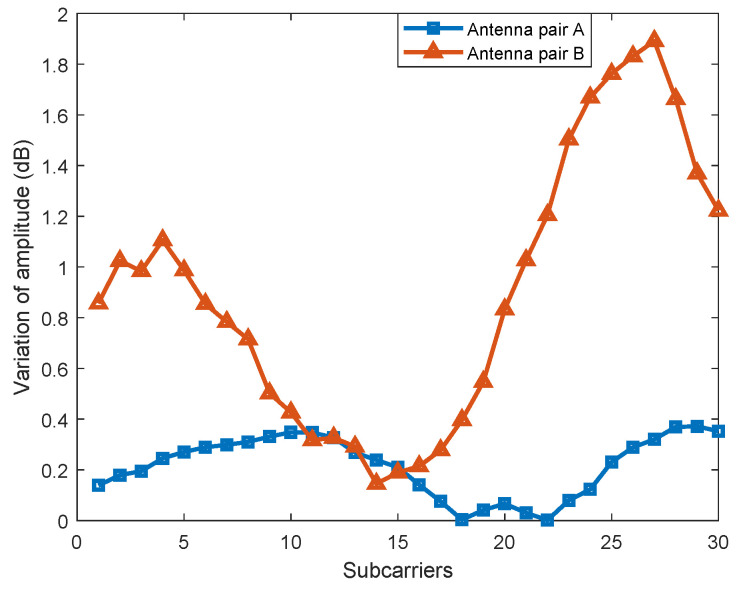
Variation of amplitude for different antenna pairs from LA to LB.

**Figure 5 sensors-23-05830-f005:**
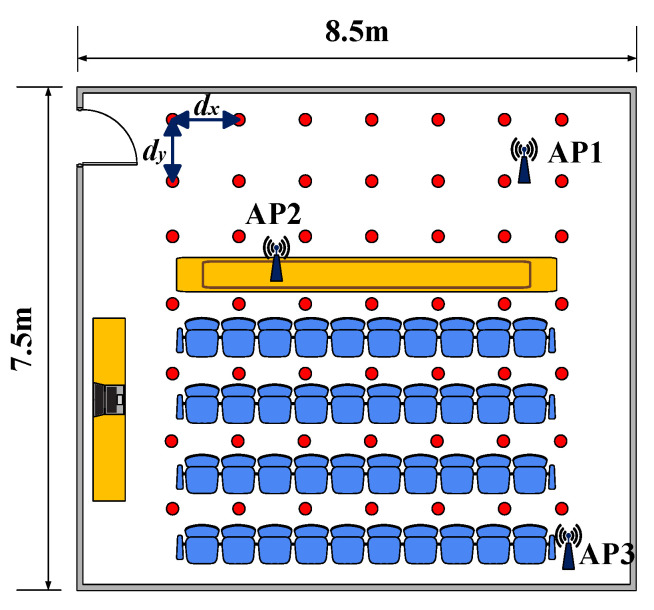
The layout of the conference room.

**Figure 6 sensors-23-05830-f006:**
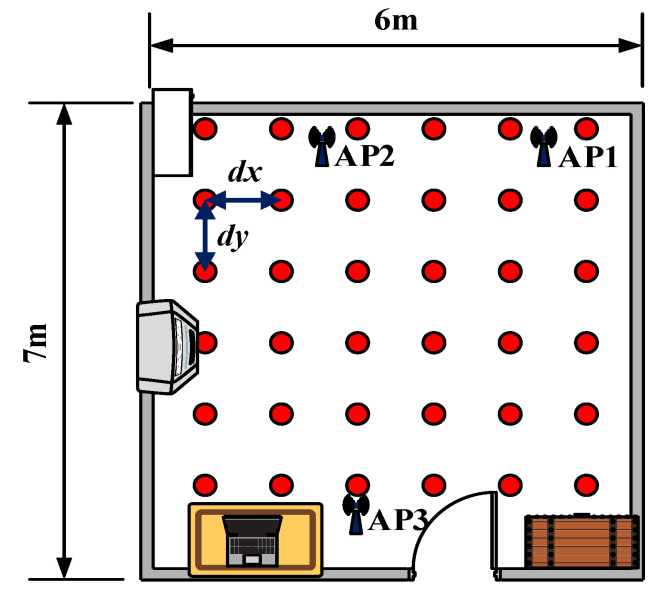
The layout of the living room.

**Figure 7 sensors-23-05830-f007:**
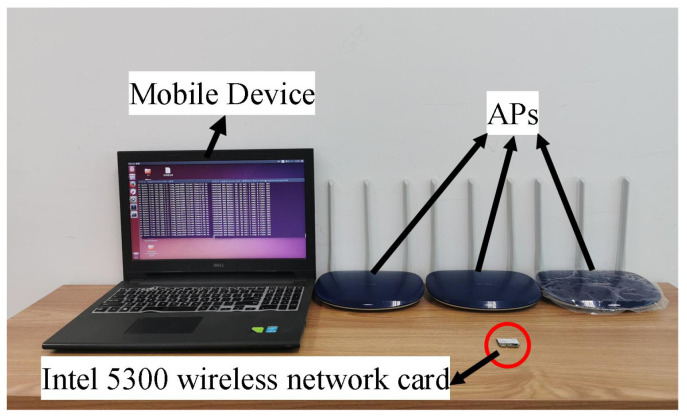
The experimental system.

**Figure 8 sensors-23-05830-f008:**
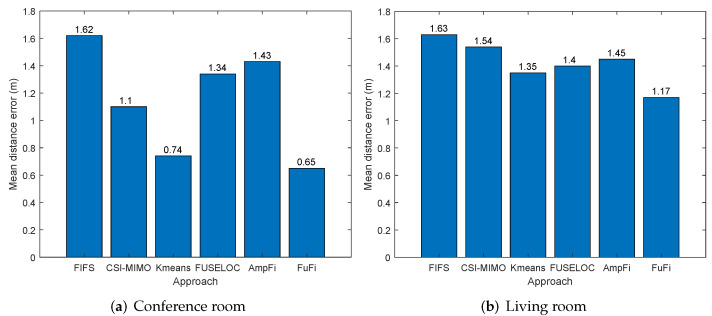
MDE of different localization approaches in the single-AP case.

**Figure 9 sensors-23-05830-f009:**
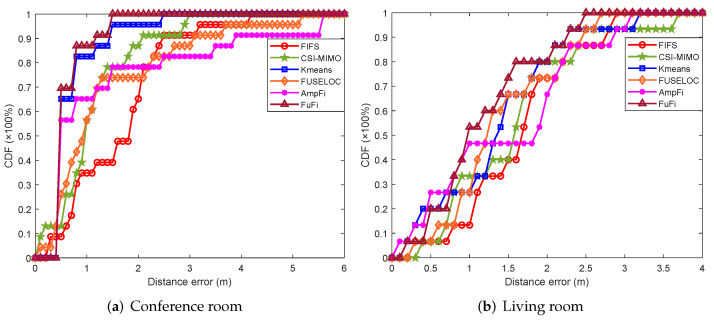
CDFs of localization error of different localization approaches in the single-AP case.

**Figure 10 sensors-23-05830-f010:**
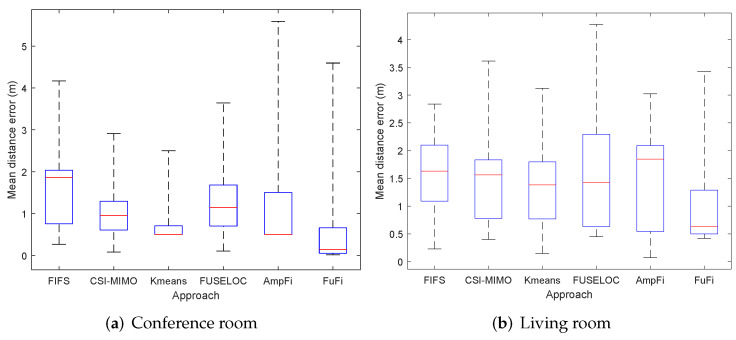
Localization error of different localization approaches in the single-AP case.

**Figure 11 sensors-23-05830-f011:**
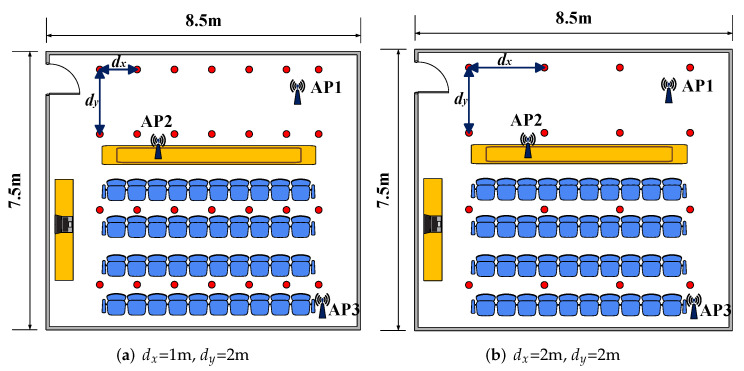
The layout of the reference points in the conference room.

**Figure 12 sensors-23-05830-f012:**
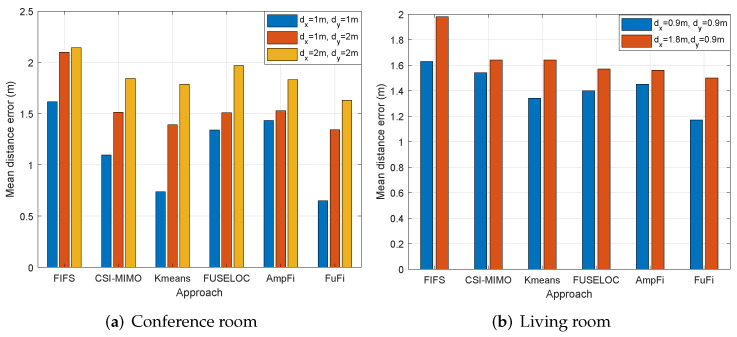
MDEs for different localization approaches with different reference point intervals.

**Figure 13 sensors-23-05830-f013:**
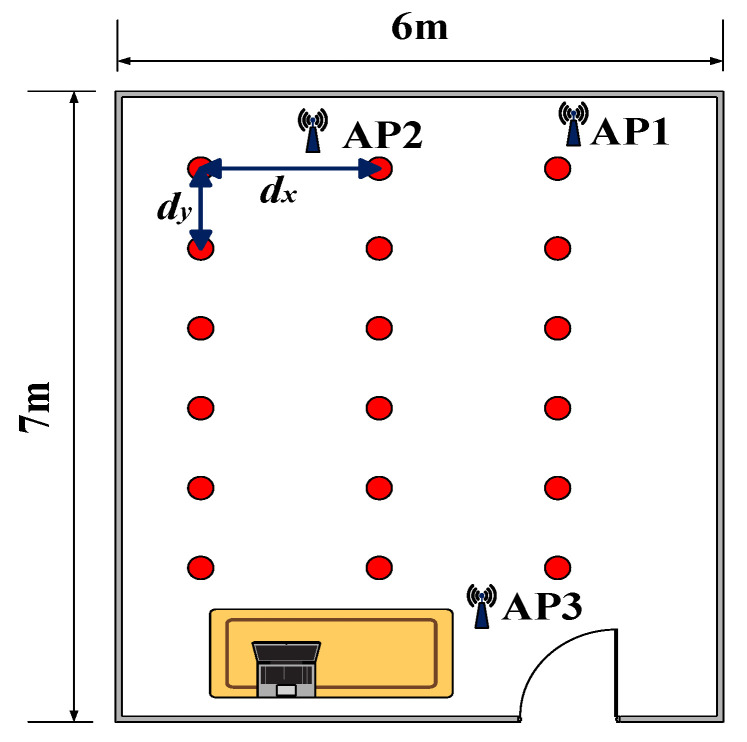
The layout of the conference points in the living room.

**Figure 14 sensors-23-05830-f014:**
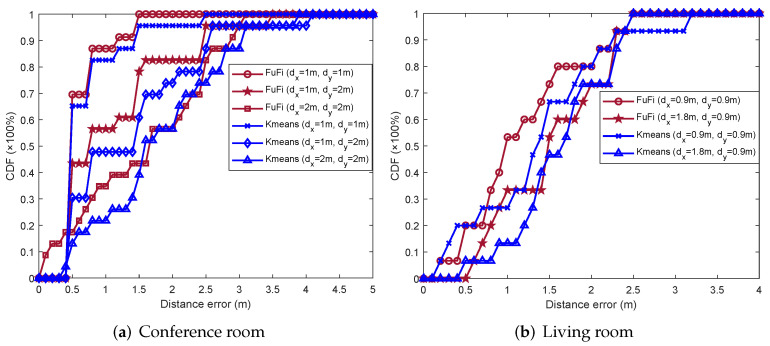
CDFs of error for different localization approaches with different reference point intervals.

**Figure 15 sensors-23-05830-f015:**
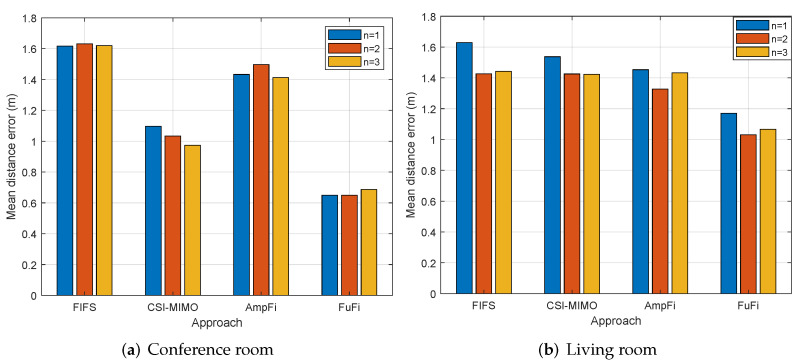
MDEs for different localization approaches with different numbers of reference points.

## Data Availability

No new data were created or analyzed in this study. Data sharing is not applicable to this article.

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
