# Peer review of "Channel State Information Based Indoor Fingerprinting Localization"

_sensors, 2023, doi:10.3390/s23135830_

Round 1

Reviewer 1 Report

In order to achieve indoor positioning, the article selects the WIFI channel state information (CSI) with fine-grained characteristics as the feature to distinguish the location. However, the AmpFi and FuFi proposed in this paper only deal with the amplitude of the subcarrier with a relatively simple algorithm, and the degree of innovation of this article is open to question.

Point1:

What is the basis for the selection of K and Tdis in article 4.4.2?

Point 2:

How does the indoor environment change (such as the impact of pedestrians passing by, unknown changes in indoor objects, etc.) affect the offline and online stages? How does it affect the final positioning accuracy?

Point 3:

Compared with AmpFi, FuFi mentioned in the article uses full-dimensional channel information to improve system accuracy. However, what is the impact of this scheme on the amount of calculation in the online phase?

Point 4:

How often does the scheme need to update the fingerprint database data in the actual long-term positioning to reduce errors? Does it have better performance in this respect compared to RSSI signal acquisition?

Minor editing of English language required.

Author Response

We would like to thank you for your careful reading, helpful comments, and constructive suggestions, which has significantly improved the presentation of our manuscript.

Q1: What is the basis for the selection of K and Tdis in article 4.4.2?

Response: Thanks for your valuable counsel. According to your suggestion, we have added some modifications to explain the determination of K and Tdis in the corresponding part. The values of K and Tdis are obtained by extensive experiments in labs. Indoor environments are complex and uncertain, and the values of K and Tdis are taken differently in order to optimize localization performance. Therefore, in different indoor environments, we determine the optimal values of K and Tdis in the current indoor environment by extensive experiments in the offline phase.

Q2: How does the indoor environment change (such as the impact of pedestrians passing by, unknown changes in indoor objects, etc.) affect the offline and online stages? How does it affect the final positioning accuracy?

Response: Thanks for your valuable counsel. In this paper, the proposed approaches can provide accurate localization service in the indoor environment that remains unchanged in both the online phase and the offline phase. If the indoor environment of the online phase changes compared with that of the offline phase, the positioning accuracy may be affected. In this paper, we consider that high precision indoor location service can be realized under the condition of the constant indoor environment. In order to solve your concerns, we will consider adding transfer learning in the future work to improve the robustness of the system.

Q3: Compared with AmpFi, FuFi mentioned in the article uses full-dimensional channel information to improve system accuracy. However, what is the impact of this scheme on the amount of calculation in the online phase?

Response: Thanks for your valuable counsel. Compared with AmpFi, FuFi adopts a higher dimensional fingerprint which contains more information about the indoor environment, but inevitably, the computational effort increases. Fortunately, positioning accuracy will be significantly improved while bringing a millimeter delay that can be ignored. Therefore, the impact of this scheme on the increased amount of calculation in the online phase can be negligible especially when the accuracy has been greatly improved.

Q4: How often does the scheme need to update the fingerprint database data in the actual long-term positioning to reduce errors? Does it have better performance in this respect compared to RSSI signal acquisition?

 Response: Thanks for your valuable counsel. If the indoor environment does not change significantly, the fingerprint database is unnecessary to be updated. The information contained in CSI is more fine-grained than that contained in RSSI, which can better reflect the indoor environment, and the positioning accuracy is higher using the CSI positioning method.

Thank you again for your positive and constructive comments and suggestions on our manuscript. We hope you will find our revised manuscript acceptable for publication.

Reviewer 2 Report

1.       The title of the paper is not so impressive, change it as per your work.

2.       From the initial round how one can write with different references.

3.       How LSTM is useful for your data, explain it with appropriate reason.

4.       What is the authenticity of your data and how you collected it.

5.       What are different pre-processing method you have used.

6.       Why you used here the Euclidian distance not the manhating one.

7.         

1.       The title of the paper is not so impressive, change it as per your work.

2.       From the initial round how one can write with different references.

3.       How LSTM is useful for your data, explain it with appropriate reason.

4.       What is the authenticity of your data and how you collected it.

5.       What are different pre-processing method you have used.

6.       Why you used here the Euclidian distance not the manhating one.

7.         

Author Response

We would like to thank you for your careful reading, helpful comments, and constructive suggestions, which has significantly improved the presentation of our manuscript.

Q1: The title of the paper is not so impressive, change it as per your work.

Response:Thanks for your valuable counsel. As you suggested, we changed the title of this paper to “Channel State Information Based Indoor Fingerprinting Localization” to make it more impressive and in tune with the theme.

Q2: From the initial round how one can write with different references.

Response: Thanks for your valuable counsel. Reference points should be randomly selected and distributed as evenly as possible in the environment to be positioned. After that, users should stand at different reference points successively and collect corresponding CSI data.

Q3: How LSTM is useful for your data, explain it with appropriate reason.

Response:Thanks for your valuable counsel. Actually, LSTM is not used in this paper.

Q4: What is the authenticity of your data and how you collected it.

Response: Thanks for your valuable counsel. According to your suggestion, we modified our manuscript in the evaluation part to explain the source and process of data collection. The experimental data was actually collected in our two different laboratories. Specifically, three APs and a laptop were placed in fixed positions in our labs when collecting data. In the offline phase, the users were asked to stand on the reference points successively, and the CSI was collected when the users stood on different reference points. In the online stage, the user stands at the test point randomly, and the CSI of the user is collected for positioning.

Q5: What are different pre-processing method you have used.

Response: Thanks for your valuable counsel. There are three main differences which are listed as follows:

(1)   The full-dimensional CSI amplitude is taken as a fingerprint to enrich the data volume of the fingerprint and reflect the indoor environment more comprehensively.

(2)   The full-dimension amplitude is normalized so that the CSI of each transmission link can participate in positioning fairly and reduce the influence of CSI amplitude between different transmission links on positioning. For example: If the average amplitude of link 1 is 100db, the change of 10% is 10db, while the average amplitude of link 2 is 20db, and the change of 20% is only 4db, its influence in positioning will be less than that of link 1, but obviously, link 2 is more sensitive to indoor environment changes.

(3)   IWKNN algorithm can eliminate outliers in the nearest neighbor reference points, which is helpful to improve the positioning accuracy.

Q6: Why you used here the Euclidian distance not the manhating one.

Response: Thanks for your valuable counsel. Euclidean distance as a method of measuring the distance between two points has many advantages, including:

(1)   Intuitive and easy to understand: Euclidean distance is one of the most common distance measurement methods, simple calculation, intuitive, and easy to understand.

(2)   Good mathematical properties: Euclidean distance satisfies the basic properties of distance, such as non-negativity, symmetry, and triangle inequality.

(3)   Widely used: Euclidean distance is widely used in many fields, such as machine learning, data mining, image processing, and so on.

(4)   Strong applicability: Euclidean distance is applicable to continuous data, such as numerical data, time series, etc.

In short, Euclidean distance has the advantages of intuitiveness, good mathematical properties, wide application, and strong applicability. Hence, we adopt the Euclidian distance in this paper.

Thank you again for your positive and constructive comments and suggestions on our manuscript. We hope you will find our revised manuscript acceptable for publication.

Reviewer 3 Report

The amplitude of CSI-based indoor fingerprinting localization (AmpFi) and full-dimensional CSI-based indoor fingerprinting localization (FuFi) are the two indoor localization methods that the authors of this research suggest using Wi-Fi's channel state information (CSI). AmpFi proposes the improved weighted K-nearest neighbour (IWKNN) to estimate the unknown locations in the online phase and uses the amplitude of CSI as the localization fingerprint in the offline phase. On the basis of AmpFi, FuFi is developed, which uses the normalised amplitudes of the full-dimensional subcarriers as a fingerprint and treats each subcarrier in the MIMO system as an independent feature. On the commercial network interface card (NIC), AmpFi and FuFi are both integrated, and FuFi works better than a number of other common fingerprinting-based indoor localization techniques.

The article fits well with the content of the journal "Sensors", the section "Internet of Things" and the special feature "Indoor Positioning Technologies for Internet-of-Things".

The study offers a thorough analysis of related work on Wi-Fi indoor localisation, including fingerprinting- and link-based methods. The benefits of CSI over received signal strength (RSS) for indoor localisation are also discussed by the authors. The offline and online stages of the proposed AmpFi and FuFi techniques, as well as the enhanced weighted K-nearest neighbour algorithm, are all detailed in depth. In two distinct situations, a living room and a conference room, the testing findings demonstrate that FuFi beats AmpFi and many other common fingerprinting-based indoor localization systems.

The paper is well-written overall and makes a significant addition to the topic of indoor Wi-Fi localisation. The experimental findings show the efficiency of the suggested procedures, which are novel. However, here are some ideas to enhance the article's quality:

1.       Give more information about the experimental setup: It would be beneficial to offer additional details regarding the precise setup utilised in the living room and conference room environments in order to have a better understanding of the tests that were carried out. The repeatability of the studies would be improved by providing information on the rooms' dimensions, the quantity and placement of access points, and any potential sources of interference.

2.       Explain the assessment metrics: Although it is stated that FuFi performs better than AmpFi and other methods, it would be advantageous to add more information on the evaluation measurements itself, such as localization accuracy, precision, or error rates. The readers would be better able to comprehend the comparative benefits of FuFi if these measurements were provided.

3.       Discuss the techniques' shortcomings and future research: It would be beneficial to have a section that does so. A more complete viewpoint would be provided by addressing potential obstacles or limitations, such as the impact of environmental variables or scaling problems. The importance of the study would also be increased by making recommendations for prospective future research directions or enhancements to the suggested methodologies.

The authors may enrich the article and raise the level of quality by resolving these issues.

Author Response

We would like to thank you for your careful reading, helpful comments, and constructive suggestions, which has significantly improved the presentation of our manuscript.

Q1: Give more information about the experimental setup: It would be beneficial to offer additional details regarding the precise setup utilised in the living room and conference room environments in order to have a better understanding of the tests that were carried out. The repeatability of the studies would be improved by providing information on the rooms' dimensions, the quantity and placement of access points, and any potential sources of interference.

Response:We feel great thanks for your professional review work on our paper. According to your nice suggestions, we have modified our manuscript in the evaluation part to provide more information about the experimental setup. Information parameters related to the experiment, including room size, number of access points and location, have been marked and described in the experimental diagram in this paper. Besides, there may be some possible sources that can interfere the positioning accuracy. And they can be indoor household appliances, walking passers-by, changes in the position of visible objects, etc.

Q2: Explain the assessment metrics: Although it is stated that FuFi performs better than AmpFi and other methods, it would be advantageous to add more information on the evaluation measurements itself, such as localization accuracy, precision, or error rates. The readers would be better able to comprehend the comparative benefits of FuFi if these measurements were provided.

Response:Thanks for your valuable counsel. One of the most important indicators of positioning is positioning accuracy, which can reflect whether a positioning method can meet the given accuracy requirements to meet the needs of users. Therefore, extensive experimental results are provided in this paper to demonstrate that the proposed approaches possess high positioning accuracy. However, in location-based services, there is usually no definition of accuracy, error rate, etc. For example, the positioning result of a positioning algorithm is quite different from the original position. We can only say that the positioning accuracy of the algorithm is poor, but it is not proper to say that the algorithm is inaccurate or its error rate is high.

Q3: Discuss the techniques' shortcomings and future research: It would be beneficial to have a section that does so. A more complete viewpoint would be provided by addressing potential obstacles or limitations, such as the impact of environmental variables or scaling problems. The importance of the study would also be increased by making recommendations for prospective future research directions or enhancements to the suggested methodologies.

Response:Thanks for your valuable counsel. The disadvantage of this study is that the indoor environment of the offline stage and online stage needs to remain unchanged. Future research direction can use machine learning and other methods to collect only a small amount of reference point data after the environment changes in the indoor environment of the online stage, and use these small amounts of data to update the entire offline fingerprint database. Reduce workload and provide system robustness and adaptability.

Thank you again for your positive and constructive comments and suggestions on our manuscript. We hope you will find our revised manuscript acceptable for publication.

Round 2

Reviewer 1 Report

1. The assumption based on this article is that when the indoor environment is relatively stable, however, meeting rooms, living rooms and other places have a large flow of people, and the influence of factors such as people walking on the signal cannot be ignored, which limits the application scenarios of this article.

2. In part 4.4.2 of the article, the author mentioned that T and Tdis are obtained from a large amount of experimental data, and indoor environments are complex and uncertain. In order to optimize the positioning performance, the values of K and Tdis are also different. These environment-dependent parameters also lead to application limitations.

Author Response

We would like to thank you again for your constructive suggestions and rigorous comments.

Q1: The assumption based on this article is that when the indoor environment is relatively stable, however, meeting rooms, living rooms and other places have a large flow of people, and the influence of factors such as people walking on the signal cannot be ignored, which limits the application scenarios of this article.

Response: Thanks for your valuable counsel and for pointing out this problem. Your comments are very valuable. In many scenarios, such as meeting rooms, there is a high flow of people, which can affect positioning accuracy. As you said, the assumption of a stable indoor environment is very important to achieve accurate indoor location. Hence, this paper mainly aims at those scenarios where the environment is relatively stable. For example, the proposed indoor localization approaches can play a significant role in applications with confined spaces, such as tunnels and mines, which are introduced in our manuscript. Therefore, the proposed approaches can be deployed to provide accurate indoor location services in applications where the environment is relatively stable.

Q2: In part 4.4.2 of the article, the author mentioned that T and Tdis are obtained from a large amount of experimental data, and indoor environments are complex and uncertain. In order to optimize the positioning performance, the values of K and Tdis are also different. These environment-dependent parameters also lead to application limitations.

Response: Thanks for your valuable counsel, and the problem you pointed out is very constructive. Although the proposed approaches have some limitations in scenarios where the environment is changeable, they can make a big difference in some applications where the environment is relatively stable. As mentioned in the previous question, in some applications with confined spaces, such as tunnels and mines, the proposed approaches are urgently needed to achieve accurate indoor positioning. Among these applications, the parameters remain almost constant, and the fingerprint database also doesn't need to be updated frequently. Besides, the approaches can overcome the drawback that GPS is unable to complete accurate indoor localization.

Thank you again for your positive and constructive comments and suggestions on our manuscript. We hope you will find our revised manuscript acceptable for publication.
